# ‘De Novo’ Brain AVMs—Hypotheses for Development and a Systematic Review of Reported Cases

**DOI:** 10.3390/medicina57030201

**Published:** 2021-02-26

**Authors:** Ioan Alexandru Florian, Lehel Beni, Vlad Moisoiu, Teodora Larisa Timis, Ioan Stefan Florian, Adrian Balașa, Ioana Berindan-Neagoe

**Affiliations:** 1Clinic of Neurosurgery, Cluj County Emergency Clinical Hospital, 400012 Cluj-Napoca, Romania; lehelbeni@yahoo.com (L.B.); vlad.moisoiu@gmail.com (V.M.); stefanfloriannch@gmail.com (I.S.F.); 2Department of Neurosurgery, Iuliu Hatieganu University of Medicine and Pharmacy, 400012 Cluj-Napoca, Romania; 3Department of Physiology, Iuliu Hatieganu University of Medicine and Pharmacy, 400006 Cluj-Napoca, Romania; doratimis@gmail.com; 4Clinic of Neurosurgery, Tîrgu Mureș County Clinical Emergency Hospital, 540136 Tîrgu Mureș, Romania; adrian.balasa@yahoo.fr; 5Department of Neurosurgery, Tîrgu Mureș University of Medicine, Pharmacy, Science and Technology, 540139 Tîrgu Mureș, Romania; 6Research Center for Functional Genomics, Biomedicine, and Translational Medicine, Institute of Doctoral Studies, “Iuliu Hatieganu” University of Medicine and Pharmacy, 400012 Cluj-Napoca, Romania; ioananeagoe29@gmail.com

**Keywords:** arteriovenous malformation, de novo, acquired, seizure, hemorrhage

## Abstract

*Background and Objectives:* Brain arteriovenous malformations AVMs have been consistently regarded as congenital malformations of the cerebral vasculature. However, recent case reports describing “de novo AVMs” have sparked a growing debate on the nature of these lesions. *Materials and Methods:* We have performed a systematic review of the literature concerning de novo AVMs utilizing the PubMed and Google Academic databases. Termes used in the search were “AVM,” “arteriovenous,” “de novo,” and “acquired,” in all possible combinations. *Results:* 53 articles including a total of 58 patients harboring allegedly acquired AVMs were identified by researching the literature. Of these, 32 were male (55.17%), and 25 were female (43.10%). Mean age at de novo AVM diagnosis was 27.833 years (standard deviation (SD) of 21.215 years and a 95% confidence interval (CI) of 22.3 to 33.3). Most de novo AVMs were managed via microsurgical resection (20 out of 58, 34.48%), followed by radiosurgery and conservative treatment for 11 patients (18.97%) each, endovascular embolization combined with resection for five patients (8.62%), and embolization alone for three (5.17%), the remaining eight cases (13.79%) having an unspecified therapy. *Conclusions:* Increasing evidence suggests that some of the AVMs discovered develop some time after birth. We are still a long way from finally elucidating their true nature, though there is reason to believe that they can also appear after birth. Thus, we reason that the de novo AVMs are the result of a ‘second hit’ of a variable type, such as a previous intracranial hemorrhage or vascular pathology. The congenital or acquired characteristic of AVMs may have a tremendous impact on prognosis, risk of hemorrhage, and short and long-term management.

## 1. Introduction

Arteriovenous malformations (AVMs) of the brain consist of feeder arteries that amalgamate and form a nidus, shunting the oxygenated blood directly into the venous system without an interposing capillary network [1,2,3,4]. The most common and undeniably the most severe clinical presentation is intracranial hemorrhage, AVMs representing an important source of mortality and morbidity especially in the younger population. Epilepsy, progressive neurological deficit, and chronic headache are less frequently encountered in patients harboring this pathology. Asymptomatic patients may benefit from simple clinical and imagistic observation, whereas those with minimal symptoms can be managed conservatively. Currently the most effective and permanent treatment method or ruptured or symptomatic AVMs consists of microsurgical resection of the nidus and rigorous hemostasis, but radiosurgery and endovascular embolization are also viable management options. 

AVMs are traditionally considered congenital, an estimated 95% of these lesions being sporadic in nature, whereas the remaining 5% are familial and most often associated with hereditary hemorrhagic telangiectasia (HHT) [4]. The congenital aspect derives from the observation that these lesions may occur at any age, from infants and children to adults and elderly alike, even though there is a scarcity of conclusive data to demonstrate this fact [5,6,7]. 

Our current grasp of cerebral AVMs has dramatically improved over the last few decades, by delving into the genetic and epigenetic intricacies of their pathogenesis [8,9,10]. Data concerning the molecular underlying of AVMs come from AVM syndromes as well as from the analysis of sporadic cases of AVMs. For instance, genetic events involving members of the transforming growth factor-β (TGF-β) signaling, namely Endoglin (ENG) and Activin A receptor type I (ACVR1 or ALK1) are causative in HHT type 1 and type 2, respectively [11,12]. Mutations in SMAD4, which is a downstream effector of TGF-β, causes combined syndrome of HHT with juvenile polyposis [13]. In agreement with such an interpretation, conditional knock out of either ENG or ALK1 resulted in arteriovenous malformations of the skin in mice only after a secondary injury, for example wounding [14,15]. Moreover, vascular endothelial growth factor (VEGF) has the ability to mimic this wounding effect on skin AVM development, at least in animal models [16]. In the case of the more common sporadically occurring AVMs, somatic KRAS mutations were discovered in approximately 55% of brain AVM specimens, also linked to a lower patient age, and a larger nidus mean size [17,18,19], whereas the non-coding polymorphism in ALK1 (ALK1 IVS3-35A>G) was established as a genetic risk factor in two independent studies [20,21]. Other polymorphisms linked to sporadic AVMs include polymorphisms in ENG [20], interleukin-1β (IL1B) [22] and VEGF [23]. Even so, we are currently unable to effectively validate whether these malformations actually develop during embryogenesis and evolve during one’s lifetime, or if they appear at a later point in time when certain microenvironmental conditions are met.

There is a mounting number of pertinent case reports detailing AVMs that occurred sporadically after credible evidence of their initial absence. Either adequate imaging studies or intraoperative examinations preceded the certain appearance of these vascular malformations, which were then discovered at a variable interval. A diversity of preexisting circumstances or pathologies either influenced or initiated the development of AVMs in already susceptible patients. Ischemic stroke and transient ischemic attacks (TIA) were among the most common ailments preceding de novo AVMs [24], as were seizures [25]. Certain individuals previously harbored another type of intracranial vascular pathology, or even a differently located AVM, highlighting the irrefutable predisposition for such an occurrence [26]. Intracranial tumors with a highly vascular component have also been described as potential triggers for previously absent AVMs [27]. An earlier hemorrhagic episode [28] and even head trauma of variable severity have also been reported as generators for de novo AVMs [29].

Notwithstanding the fact that cerebral AVMs are a rare pathology, they are cited as the most common cause of intracranial hemorrhage in children and young adults, leading to a lifelong threat of death or disability [2,3,4,30,31]. As such, elucidating their true natural history is crucial in order to establish the best possible course of diagnosis and management, and help lessen their socio-economic encumberment. In this systematic review we have included, to the best of our knowledge, all such incidences reported in the available and relevant literature. 

## 2. Material and Methods

We performed a rigorous examination of the PubMed and Google Academic medical databases utilizing the terms “AVM”, “arteriovenous malformation”, “brain”, “cerebral”, “intracranial”, “acquired”, and “de novo”, and all of their possible combinations. Our search only identified articles in English, and we excluded animal studies from the systematic review. Other systematic reviews were also discarded. Afterwards, we methodically assessed the references of the manuscripts gathered for any additional articles significant to our research. Among acquired AVMs that appeared after a previous pathology or injury, we have also included AVMs that recurred at a different site after a previous angiographically-demonstrated occlusion. We paid special attention not to include data from overlapping series that were published at a different time or in another journal. Nevertheless, certain redundancies cannot be completely dismissed.

Microsoft^®^ Excel for Mac, version 16.37, was used to create a spreadsheet wherein all relevant cases were gathered in order to create our database. The same program was utilized to perform descriptive statistics, as well as to create the appropriate graphs.

## 3. Results

### 3.1. Literature Data

As of June 2020, our search yielded 218 results on PubMed. After careful consideration of titles and abstracts, we have eliminated 163 articles which were not relevant to the subject at hand, followed by the exclusion of an additional 16 that were either in vitro or animal studies, other systematic reviews, case reports detailing other vascular pathologies such as arteriovenous fistulas, aneurysms, cavernomas, or AVMs outside of the cranial cavity. Among these, two articles which represented correspondence regarding the same article, as well as a letter to the editor concerning another article, were eliminated. Moreover, 19 further articles were identified subsequent to the scrupulous reference review of the previously retrieved studies. As such, we have obtained a total of 58 cases of presumed de novo AVMs from 53 articles matching our inclusion criteria. We then divided the patients according to pathological subgroup of the presumed predisposing disease, and in alphabetical order of the first author, as seen in Table 1. The flow-chart for our literature gathering is summarized in Figure 1.

### 3.2. Demographic Data and Descriptive Statistics

We identified 58 patients harboring de novo AVMs (AVMs that appeared on subsequent imaging studies after a definitive negative one), out of which 32 were male (55.17%), and 25 were female (43.10%). Mean age at de novo AVM diagnosis was 27.833 years (standard deviation (SD) of 21.215 years and a 95% confidence interval (CI) of 22.3 to 33.3), with the extremes being 2.5 years [32] and 80 years [26], respectively (Figure 2). The gender and age of one patient (1.72%) was unspecified in the respective case report [33] and was therefore not included when calculating mean age and standard deviation.

Whereas original investigations varied between CT (either contrast-enhanced or otherwise), MRI (including magnetic resonance venography), tran-sfontanellar ultrasonography (US) in one case, or DSA, or any combination of these, all purportedly de novo AVMs were subsequently confirmed via angiography, with the exception of two (3.44%) that only had MRI scans [34,35] or CT scans [1,36]. The time from initial imaging study, which did not reveal any AVMs, and the diagnostic study varied from three months [37] to 25 years [36], the average being 7.67 years and with a SD of 6.008 years (95% CI of 6.11 to 9.23).

We grouped the patients according to the major underlying pathology allegedly associated with the development of de novo AVMs. We considered ten major categories, in descending order of frequency: preexisting cerebrovascular malformations (including other AVMs, cavernomas and aneurysms)—13 patients (22.41%); hemorrhagic stroke—11 cases (20.69%); epileptic seizures—nine individuals (15.52%); brain tumors—six patients (10.34%); ischemic stroke—five patients (8.62%); Moyamoya disease—four cases (6.9%); traumatic brain injury—three patients (5.17%); and genetic syndromes, inflammatory diseases, and liver cirrhosis, each with two individuals (3.45%). This is also illustrated in Figure 3.

Once diagnosed, the majority of de novo AVMs were treated via microsurgical resection (20 out of 58, or 34.48%), followed by radiosurgery and conservative management for 11 patients (18.97%) each (Figure 4). Endovascular embolization combined with subsequent resection was the choice method of therapy for five patients (8.62%), whereas only three (5.17%) underwent embolization alone. For the other eight cases (13.79%), therapy was not mentioned within their respective articles.

Half of the patients had a favorable outcome after treatment, either being neurologically intact, or recovering partially or completely. The natural history of five (8.62%) culminated with death, the shortest period being only six days after surgery due to massive pulmonary embolism [38], and the longest at 3.5 years after diagnosis and endovascular embolization of the AVM [32]. The outcome was not specified for the remaining 24 cases (41.38%).

## 4. Discussion

### 4.1. Data Review

As evidenced by the demographic data, the majority of patients reported with de novo AVMs (64.91%) were under the age of 30 at the time of diagnosis, highlighting that even under the circumstances of acquired vascular malformations, this remains a pathology of the young. Male-to-female ratio was 1.28, with a slight yet statistically insignificant predominance in the male gender, which corresponds to the patient populations described in the majority of reported series [2,3,4,30,31]. The interval between the initial investigation and definitive identification of acquired AVMs varied widely, the shortest being only three months in a patient who harbored an anaplastic astrocytoma immediately adjacent to the AVM itself [37], and the lengthiest reaching 25 years in a patient who had been treated with phenytoin for epileptic seizures during this time [36]. However, we point out that in the report of Ozsarac et al., the initial angiographic studies were not available, and it was the patient himself who stated upon recollection that those investigations had been normal. As such, it is difficult to properly ascertain the moment when the AVM appeared.

It may be worthwhile mentioning that some critics believe de novo AVMs are in fact occult vascular lesions, only later visible as the result of low sensitivity and specificity of certain imaging techniques, specifically DSA [29,39]. Even so, the majority of case reports illustrating de novo AVMs have been published within the last two decades, during an ever-increasing accessibility to high-resolution and, serial cross-sectional imaging techniques, as well as an upsurge in detection of once-occult intracranial lesions. 

In the following section, we will discuss the cases according to the underlying pathologies and emit a hypothesis on their probably pathogenesis. Due to the rare nature of acquired AVMs and, hence of their pathogenesis, the current theories regarding congenital AVM development may not apply here. 

### 4.2. Finding the Culprit—Hypothesized Causes for De Novo AVMs

#### 4.2.1. Intracranial Vascular Malformations and Aneurysms

The majority of the reported patients included in this systematic review had a previous cerebral vascular malformation or degenerative lesion before the occurrence of a de novo AVM (13, or 22.41%). These ranged from dural arteriovenous fistulas (dAVFs), preexisting AVMs or cavernous malformations, or even aneurysms. Patients with Moyamoya disease are covered in a separate section, due to its frequency and implications. Among these patients, seven (12.07%) had a previous AVM that was treated either invasively or conservatively, the acquired lesion appearing either in the same location, or separately [12,26,40,41,42]. Nagm et al. suspected that the occurrence of de novo AVMs was the result of certain environmental stimuli that triggered angiogenesis via the upregulation of TGF-β and VEGF [43]. They also entertained the concept of increased immunoactivity processes, since the analysis of the malformation samples acquired from the second intervention in their patient revealed heightened levels of ENG in the perivascular area, as well as accentuated immunoreactivity for phosphorylated extracellular signal-regulated kinase (pERK).

In three of the patients, it was hypothesized that entrance radiation originating from stereotactic radiosurgery (SRS)-treated vascular lesions either facilitated or triggered the development of de novo AVMs [26,41,43]. After the removal of the recurrent nidus, Kawashima et al. discovered a slightly increased Ki-67 expression, in conjunction with negative expression of both ENG and VEGF, whereas raised ENG expression was noticed in the area surrounding the nidus and exhibiting radiation-induced inflammation [41]. Another prospect is that, in the scenario of a recurring AVM, the recanalization of thrombosed vessels can ensue at some point in time after radiosurgery [43]. Correspondingly, the blood flow redistribution ensuing gradual obliteration of vascular malformations may expedite immature vessels around the original lesion, generating a secondary AVM. Therapeutic exposure to radiation in various pathologies has been demonstrated to generate endothelial cell injury, thus prompting vascular remodeling and VEGF and TGF-β production [24,26,44,45,46,47,48]. In addition, radiation itself may produce mutations in the genes involved in AVM development. It is therefore likely that radiation indeed plays as a veritable ‘second hit’ in the generation of de novo AVMs.

Prior to the discovery of acquired AVMs, three of the cases presented suffered from cavernomas, all of which were situated in different parts of the brain than the secondary malformation [27,49,50]. While cavernomas and AVMs possess distinct genetic backgrounds in their pathogenesis [51,52], these cases may suggest that there might be a common origin within the signaling pathway of these vascular malformations, supplementing the already enthralling complexity of intracranial AVMs [50]. Perhaps a less alluring but nonetheless credible theory is that behind these de novo lesions lies an angiogenically dynamic microenvironment stemming from a mixture of insults brought on by surgical trauma and cerebral hemorrhage [28].

Disturbances within the cerebral venous drainage may also play a part in the genesis of de novo AVMs, as it does for dAVFs [42]. A child and an adult, both male, harbored dAVFs and were treated via embolization and radiosurgery combined with embolization, respectively [53,54]. While it could be reasoned that both AVMs were simply missed on the initial scans, an intriguing possibility is that the local venous hypertension brought about by the occlusion of the dAVFs led to the growth of de novo AVMs.

A single case of surgically clipped unruptured intracranial aneurysm, without any other concomitant vascular malformations, was linked to an acquired AVM in the territory of that respective artery [55]. The authors argued that certain physiological stimuli such as vascular shear stress and prolonged cerebral hypoxia after aneurysmal clipping might reactivate angiogenesis. As this is an isolated incidence, other possibilities have yet to be expounded.

#### 4.2.2. Hemorrhagic Stroke

According to the information presented, hemorrhagic stroke from various sources is the single most common primary brain injury leading to the formation of de novo AVMs within the same site. Although angiographically occult fistulas masked by hematomas, thrombosis, arterial feeder vasospasm, or posthemorrhagic brain edema cannot be dismissed, a possible pathophysiological mechanism for AVM generation in a hemorrhagic cerebral environment should be postulated. It is likely that hemorrhagic stroke in conjunction with surgical trauma fosters appropriate angiogenetic conditions [28,56,57]. It may also seem likely that at least some of these AVMs stemmed from a small and previously missed angioma [58]. After surgical resection, the available empty space and probable repeated silent bleeds resulting in a progressive vascular enlargement in the hematoma cavity may also explain these occurrences [59]. Jeffree et al. conjectured that AVMs might develop not though the means of dilatation of preexisting vessels, but via the generation of brand-new vessels [60]. They proposed that the levels of certain growth factors increase upon endothelial cell exposure to shear stress, later resulting in the development of a vascular network and recruitment of new vessels within a previously occult and asymptomatic malformation. The authors also speculated an alternative, namely that the hematoma itself triggers de novo AVM formation concurrently with its own resolution.

Nevertheless, it appears that acquired AVMs may develop even remotely from the location of the initial hemorrhage. One of the patients in the series by Jeffree et al. suffered from a left parietal hematoma, followed by a right parieto-occipital de novo AVM found ten years afterward [60]. The case described by Miyasaka et al. had a right parietal hematoma eight years before three AVMs were revealed [56]. Only one grew near the hematic cavity, whereas one was situated in the frontal lobe of the same hemisphere, and the remaining one in the left occipital lobe. Four more individuals who experienced SAH of unknown origin developed de novo AVMs in various areas of the brain, any time between 2 and 23 years after the original injury [33,57,58,61]. Nakamura et al. reported a more unique instance, wherein a left frontal intraparenchymal hemorrhage was surgically evacuated, followed by the therapeutic local implantation of mesenchymal stem cells secreting the neuroprotective glucagon-like peptide-1 (GLP-1) [62]. Despite some criticism [62], the authors clearly infirmed the presence of any vascular malformations at the site of surgery [62,63]. At the three-year mark after the intervention, a ruptured AVM was diagnosed in the same region of the brain. They hypothesized that aside from VEGF, GLP-1 might also play a role in promoting angiogenesis and vessel proliferation. Validating the angiogenetic role of intracranial hematomas may be possible in the near future.

#### 4.2.3. Seizures

Several instances of epileptic convulsions with no discernible underlying causes have been associated with acquired AVMs after a variable period of time since the onset of the seizures [25,35,64,65,66,67,68]. According to numerous authors, as well as our own experience, seizures represent the second most common form of manifestation of cerebral AVMs [1,31,69]. It has been previously demonstrated that VEGF is upregulated within neural and glial cells following epileptic seizures, most likely acting as a mechanism to combat postictal neurodegeneration [25,35,67,70,71,72,73,74]. Moreover, preclinical studies have revealed a variety of beneficial effects of VEGF elevation in epilepsy and status epilepticus [75,76,77]. On the other hand, pathological upsurge of VEGF can result in a disrupted blood-brain-barrier integrity, and may also further the progression of certain diseases such as demyelinating injuries and vascular malformations [71,73,75].

One particular case was preceded by a single episode of seizures nine years prior to the discovery of an AVM, which was absent on the initial MRI scan [35]. Similarly, a seven-year-old boy diagnosed with a de novo AVM experienced febrile seizures four years prior to definitive imaging study [68]. These two cases may suggest that even a solitary episode of epileptic convulsions creates the appropriate angiogenetic conditions for AVM growth. Withal, subcortical band heterotopia (SBH), a neural migration disorder, is also associated with pharmacoresistant epilepsy [78,79,80]. A nine-year-old girl with SBH also developed an acquired AVM after three years of seizures and developmental delay [67]. Furthermore, it seems that there is no requirement for the seizures to have a tonic-clonic characteristic, as exemplified in a case by Kilbourn et al. [64]. This patient was diagnosed with hydrocephalus at an infant age, and later developed autism and absence seizures, a de novo AVM being discovered at the age of 18 years. As such, it is possible that epilepsy, in any of its forms, may play a decisive role in the appearance of acquired AVMs through an exaggerated and detrimental protective mechanism against neurodegeneration.

#### 4.2.4. Brain Tumors

Although rare instances, intracranial neoplasms with a high vascularity coexisting with AVMs have been described in the recent past [81,82,83,84]. AVMs occurring after the treatment of a previous brain tumor are also remarkably rare [27,37,44,85,86]. In the case reported by Bennet et al. of a de novo cerebellar AVM occurring after the surgical resection of a vermis hemangioblastoma, it is likely that the altered hemodynamic conditions, which stemmed from the removal of the tumor, played a crucial role in the evolution of the vascular lesion [85]. It may also seem plausible that the AVM coexisted with the hemangioblastoma but was occult beforehand. Medvedev et al. described a case in which a cerebellar AVM and a hemangioblastoma coincided, also theorizing a common ancestry linking these two pathologies [87]. A similar mechanism might also be involved in the appearance of an acquired brain AVM after recurrent meningioma surgery [27]. Likewise, malignant glial tumors foster a highly proangiogenic environment, as quantified by the enhanced production of VEGF, which could prove a crucial stimulus for a de novo AVM occurrence [37,38]. A second possible reason is that an environmental exposure gave rise to both the neoplasm and the malformation, although the incriminated factor is uncertain [38,70]. Radiosurgery has been incriminated as a probable etiologic factor for malignant brain tumors, even after therapy for AVMs, despite the risks being relatively low [88,89,90]. The exposure to radiation could also be considered a stimulus for these malformations, since tumors of the fourth ventricle (an ependymoma and a medulloblastoma, respectively) undergoing postoperative radiotherapy later developed de novo AVMs [44,86]. This might be supported by the vasculopathy and vessel remodeling following radiation therapy. Regardless of etiological considerations, the simple association with such neoplasms warrants a poor prognosis and a low short-term survival from the start [27,37,38,44,81,82,83,84,85,86,87].

#### 4.2.5. Ischemic Stroke, Venous Sinus Thrombosis, and Transient Ischemic Attacks (TIA)

Only very few cases of de novo AVMs have been correlated with ischemic events of the brain. Pabaney et al. argued that an inflammatory or ischemic lesion could have hastened the development of acquired AVMs in contrast with congenital AVMs [24]. They reasoned that the stroke acted as a ‘second-hit’ that resulted in the formation a de novo AVM. Previously, it has been demonstrated that ischemia has the potential to trigger vascular proliferation via an amplified expression of hypoxia-inducible factor-1 (HIF-1), resulting in an uninhibited vascular proliferation and the occurrence of arteriovenous shunts within a patient with genetic susceptibility and alterations regarding angiogenesis and inflammatory cascades [24,51,91]. The appearance of de novo vascular lesions could also be a response to injury mechanism after stroke [92]. This might actually be the case, as other vascular malformations (i.e., dural and pial arteriovenous fistulas) can originate from infection, inflammation, or traumatic injury, advocating the concept of environmental factors promoting angiogenesis [52,66]. Regarding venous sinus thrombosis, it may seem plausible that the increased venous pressure and subsequent hypoxia in the adjacent tissues arising from such an event leads to an enhanced angiogenic activity [14,45,66,93]. During embryological growth, venous hypertension could be the result of occlusion, stenosis or agenesis. Otherwise, cerebral ischemia itself may offer a physiological incentive for angiographically occult AVMs to enlarge and become patent [94]. It is worthwhile mentioning that among this group of de novo AVM patients, only one had an ischemic event during infancy [92], the other individuals being adults at the time of initial injury.

#### 4.2.6. Moyamoya Disease (MMD)

MMD is defined as a rare chronic occlusive pathology of the cerebral vasculature, having an unidentified etiology [95,96,97,98]. It is characterized by a series of bilateral steno-occlusive alterations in the terminal portion of the internal carotid artery (ICA) and its branches, as well as a coexisting abnormal network of collateral vessels at the base of the brain. Its presentation differs according to age, children most often manifesting epileptic seizures and ischemic events, while in adults it generally leads to hemorrhagic stroke. There have been a few instances in which AVMs preceded the development of MMD, hatching the hypothesis that the increased amount of blood traveling within the malformation itself creates a turbulent flow level of the common carotid artery bifurcation, in turn generating the hyperplasia of the intima and the ensuing collateralization and progression of an acquired form of MMD [99,100,101]. The association between MMD and ulterior de novo AVMs has been described in a small number of case reports, all of which were pediatric, potentially turning the aforementioned hypothesis on its head [95,96,97,98,99,100,101]. It is possible that the analogous biological backgrounds of these two pathologies regarding the enhanced expression of proangiogenic molecules such as VEGF, or certain inflammatory molecules including tumor necrosis factor α (TNFα), MMP and IL-6, led to the occurrence of AVMs after the progression of MMD [95]. This might be the case, as the patient described by Fujimura et al. harbored an AVM supplied by the posterior circulation, which by definition was not affected by MMD. Additionally, this patient concomitantly suffered from sickle cell disease, yet no explanation could be offered whether this condition had any influence on either the appearance of MMD or the AVM. Another plausible explanation is that MMD in these patients produced an angiogenic failure, which in turn resulted in the formation of anomalous arteriovenous shunts [96]. As Schmit et al. pointed out, it could also be stipulated that the hyperangiogenic microenvironment predominant in MMD, in conjunction with a local proangiogenic stimulus (for instance VEGF or fibroblastic growth factor—FGF) as a consequence of cerebral infarction, would have been sufficient for an AVM to develop on that precise spot [98]. The role of genetic factors, whether congenital or acquired, in the presence of prolonged hypoperfusion should also be considered [97]. Nevertheless, the probability that the AVMs coexisted with MMD but were angiographically occult at the time of initial angiography cannot be entirely ruled out.

As observed in these collection of case reports, brain infarction, and associated cerebrovascular pathologies associated, may generate sufficient stimuli for an AVM to develop postnatally, at least if the patient possesses a certain susceptibility. More research is needed in this field to precisely determine the nature of these predispositions.

#### 4.2.7. Traumatic Brain Injury (TBI)

Whether head trauma itself actually contributes to the formation of a proangiogenic environment, or its association to de novo AVMs is purely coincidental, remains to be proven. Traumatic injury has, however, been reported as an etiologic factor for arteriovenous fistulae [102,103,104]. The case described by Gonzalez et al., of a three-year-old girl who suffered a mild TBI as a result of traffic accident, developed a delayed hemorrhagic lesion that manifested through generalized epileptic seizures [29]. These seizures became increasingly difficult to control, lasting for four years before the diagnosis of a de novo AVM was established. As anteriorly discussed, both the hemorrhagic and the epileptic components could be held accountable for the appearance of an acquired AVM, although the hematoma had a left frontal placement, whereas the vascular lesion was situated in the right posterior temporal lobe. No hemorrhagic collections were identified in the vicinity of the AVM to mask this anomaly. A comparable case, in which an 11-month-old girl experienced a severe TBI with an acute subdural hematoma on the left convexity, was diagnosed with an ipsilateral de novo AVM almost 11 years later, after suffering from seizure disorder for around six years [105]. In this instance, the patient was also diagnosed with posttraumatic encephalomalacia, which could in theory act as a suitable terrain for an abnormal vasculature in a developing brain. Similarly, a young boy suffered a blunt head injury and subsequent subarachnoid hemorrhage from an unknown source, and developed an AVM in the right frontal lobe nine years after the initial incident [106]. We hypothesize that the presence of TBI resulting in intracranial hemorrhage or seizure disorder may act as a ‘second hit’ that generates a de novo AVM in patients with genetic susceptibility via VEGF upregulation (Figure 5). This theory needs to be further studied.

#### 4.2.8. Genetic Syndromes

HHT is a systemic autosomal dominant genetic syndrome with a marked predisposition to develop intracranial AVMs [6,7,11,12,46,51]. Two genes that have been linked to the development of HHT are the ENG gene, which is correlated with HHT1 as well as a higher prevalence of AVMs, and the ACVRL1 or ALK1 gene, which is associated with the HHT2 phenotype and a lower brain AVM incidence [11,12]. A single patient in the reported literature was confirmed to have HHT and developed a de novo AVM [46]. At five months of age, he underwent screening for HHT, as both his mother and older brother suffered from this syndrome and harbored a symptomatic AVM of the brain and of the spine respectively, despite being asymptomatic himself. At the age of five years, a de novo AVM became manifest and the patient underwent preoperative embolization and microsurgical resection. This case underlines the potential of these vascular malformations to appear and expand within highly vulnerable individuals, without necessarily requiring the involvement of external factors. AVMs in HHT have been shown to have a bidirectional evolution, possessing both the possibility of expansion and to regression [34]. Komiyama entertained the hypothesis that congenital brain AVMs could develop until the age of two years [93]. We, however, believe that there is no predefined interval within which these lesions materialize.

Additionally, a newborn diagnosed with hepatic hemangiomatosis and congestive heart failure suffered from a de *novo AVM* in the left cerebellopontine angle (CPA), which was treated via endovascular embolization at three and four years of age respectively [32]. Before the vascular malformation was found, MRI revealed two enhancing lesions in the left CPA and pineal region, which could have been brain hemangiomas. It was not mentioned whether or not she was tested for HHT or any other genetic syndrome, and she died because of brain hemorrhage at six-years-old. Brain AVMs and hemangioblastomas have been described in patients with diffuse neonatal hemangioma-tosis (DNH) in the past [107,108]. The authors of the respective case report proposed two possibilities: either the enlargement and progression of a coincidental AVM adjacent to the preexisting CPA hemangioma, or the conversion of said hemangioma into an AVM [32]. As there are only a finite number of individuals with diffuse neonatal hemangioma-tosis, it might prove challenging to screen them for acquired cerebral AVMs.

#### 4.2.9. Inflammatory Diseases

Only two cases of inflammatory diseases not related to the cerebral vasculature and de novo AVMs were described in the literature. The first was of an unidentified inflammation or demyelinating lesion in the brainstem [109], and the second was associated with a previously diagnosed Bell’s palsy on the same side [39]. A probable mechanism is the overexpression of VEGF as a result of the inflammatory process, similar to the mechanism explained in ischemic stroke [39,52,66]. Another explanation could be increasing hemodynamic stress due to the inflammation, leading to vascular remodeling.

#### 4.2.10. Liver Cirrhosis

Cirrhosis defines a diffuse fibrosis of the liver and the alteration of the hepatic microarchitecture into structurally anomalous nodules. Despite that, to the extent of our knowledge, only two instances of de novo brain AVMs were described in patients suffering from cirrhosis, and that the correlation between these two pathologies was poorly studied, we nevertheless consider it worthwhile mentioning [6,110]. Peripheral systemic vascular malformations are already considered a mark of advanced chronic liver disease; hence it is tantalizing to assume that these lesions and cerebral AVMs share a common set of mechanisms and etiologic factors. Among these we point out an excessive production alongside a clearance reduction of factors such as VEGF, TNF- α, IL-6, MMP-3 and MMP-9, or nitric oxide synthetase (NOS), as well as a decreased catabolism of estrogens [5,110]. To further support this assumption, Shimoda et al. illustrated the case of spontaneous AVM disappearance two years after a successful liver transplantation in a patient with alcohol-induced cirrhosis [111]. It is therefore likely that the alterations instigated by the state of venous hypertension and thrombosis, combined with a hepatic “second hit,” provides the necessary stimuli for an intracranial AVM to develop in cirrhosis. More such patients should be investigated for cerebral vascular malformations further along the course of their chronic liver disease before finally establishing this correlation.

### 4.3. Past, Present, Future, and Personal Opinions

Up until recently, AVMs were adamantly considered congenital disorders with the capacity to evolve, rupture, regress, and even recur after treatment [52]. Contemporary evidence, including the case reports incorporated in this study, suggests that we are still a long way from fully elucidating these lesions, which could actually also be acquired. In research on adult mice, Chen et al. demonstrated that the deletion of ACVRL1 within endothelial cells resulted in elevated local endothelial cell proliferation throughout brain angiogenesis, as well as the induction of de novo AVMs [112]. Similarly, Walker et al. obtained lesions resembling human AVMs in adult mice after co-injecting adenoviral vector exhibiting Cre recombinase and adeno-associated viral vectors containing VEGF within their basal ganglia, thereby producing ACVRL1 mutation [113]. In their study, Park et al. demonstrated that either physiological or environmental stimuli such as injuries, alongside the respective genetic deletion, are needed for ACVRL1-deficient blood vessels to evolve into AVMs within adult mice [15]_._ Zhu et al. also managed to trigger the formation of AVMs in adult mice, this time via CRISPR/Cas9-mediated ACVRL1 gene mutations [114]. Although other experimental models exist, it is beyond the scope of this review to present them exhaustively. It is, however, safe to assume that a number of genetic mutations, in conjunction with the appropriate environmental factors, may lead to the development of de novo AVMs.

Evidence from several of the reported case studies and series suggest that radiotherapy-related injury triggers an increased VEGF production as a response-to-injury type of adaptation, which can result in the spontaneous formation of brain vascular malformations [26,41,52,115,116,117]. While the majority of these lesions are cavernomas, it cannot be fully disproved that radiation can also induce the formation of AVMs. We believe that the adaptive changes in the cerebral vasculature occurring after stereotactic radiosurgery can also induce the exaggerated angiogenesis necessary for de novo AVMs.

Individuals in which intraparenchymal hemorrhage (IPH) were surgically removed serve as irrefutable evidence that de novo AVMs can develop in the hematic cavity, without preexisting vascular lesions in the respective regions [56,59,60,118]. We tend to consider these cases especially because the intraoperative aspect cannot be replicated entirely by imaging studies, at least with current technology.

Seizures present a dual quality, on the one hand being the second most common symptom for brain AVMs, and on the other as a possible generator for de novo AVMs [25,35,36,64,66,67,68]. Although we cannot contradict the fact that epilepsy can cause the expansion of these lesions, the possibility that a preexisting angiographically occult cerebral vascular malformation triggered the aforementioned seizures cannot be completely negated.

However, arguably the most important aspect is that all of the reported cases of de novo AVMs had an initial symptom or a preexistent underlying pathology. It is therefore tempting to believe that the actual number of such individuals is much higher, yet for various reasons, many of these patients are not investigated prior to the occurrence of AVMs. The fact that not all patients had an initial DSA might be a confounding factor, as the diagnosis of brain AVM might have been missed in this case. As the majority of these patients suffered from vascular conditions such as MMD, hemorrhagic stroke, aneurysms or other allegedly congenital vascular anomalies, the predisposition for developing AVMs was indubitably present, requiring a ‘second hit’ stimulus for AVMs to properly appear (Figure 5) [66].

One of the major limitations regarding the alteration of the ‘congenital AVM doctrine’ arises from the limited number of de novo lesions, which are mostly found in singular case reports. Additionally, the exact pathophysiological mechanisms of AVM growth are conjectured, although not fully understood. A recent consensus meeting of international neurosurgical experts debated whether AVMs should be regarded as congenital or acquired [7]. The authors all agreed that the genetic predisposition is insufficient to induce AVM formation by itself, and that a ‘second hit’ of variable nature is always required. Despite the fact that the majority of patients with brain AVMs harbor mutations of the KRAS gene [17,18], certain somatic mutations have very low frequencies within these lesions. This in itself raises the concern whether these mutations actually possess functionality in AVMs development, or whether they can be accurately quantified. Another limitation that the authors acknowledge is the hypothetical nature of this review. This stems from the fact that de novo AVMs are extremely scarce, and the data surrounding their development is reduced. However, we hope that the mechanisms proposed in this work will be more adequately studied in the near future.

Therefore, we propose that brain AVMs can be either present at birth and able to evolve or regress, or acquired but in the presence of preexisting genetic susceptibility. There needs to be an association between clinical, genetic, and animal studies in order to verify this hypothesis.

## 5. Conclusions 

Increasing evidence suggests that at least some of the AVMs discovered develop some time after birth. We are still a long way from finally elucidating their true nature, though there is reason to believe that they can also appear after a proposed ‘second hit’ during a patient’s lifetime (Figure 5). The congenital or acquired characteristic of AVMs may have a tremendous impact on prognosis, risk of hemorrhage, and short and long-term management.

This systematic review presents the reported cases of de novo brain AVMs in the current literature. Despite being a rare pathology, AVMs represent the most frequent cause of intracranial hemorrhage in young patients. Furthermore, it is possible that the real number of ruptured de novo AVMs is underrepresented, since such lesions presenting with acute hemorrhage in patients that have never undergone a previous CT or MRI scan are automatically labeled as congenital in nature. 

It must be noted that all of the diagnoses of de novo AVMs were made in conjunction with a previous associated pathology or injury. Had this not been the case, imaging studies prior to the symptomatic manifestation of the acquired AVMs would have been unnecessary, and the initial absence of these vascular lesions could not have been documented. Patients who experience seizures with or without apparent underlying causes, hemorrhagic or ischemic strokes, TBI of varying severity, other cerebrovascular malformations, or highly vascular tumors should receive regular imaging screening which includes DSA more often to highlight or exclude the formation of de novo AVMs. Further research should be conducted to more adequately ascertain the congenital or acquired characteristic of brain AVMs.

## Figures and Tables

**Figure 1 medicina-57-00201-f001:**
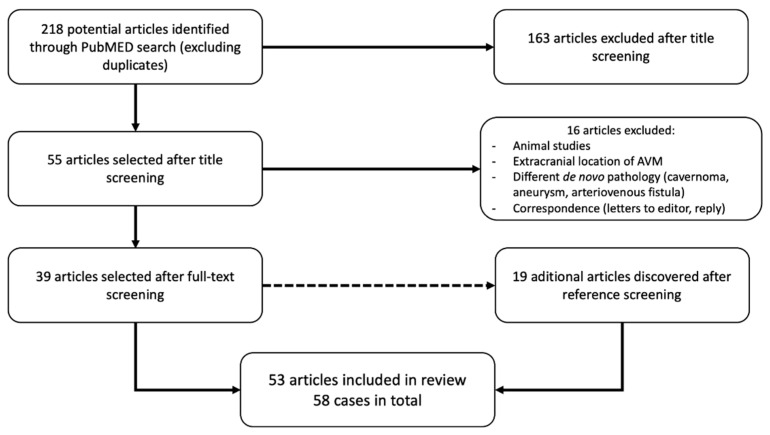
Analytical flow-chart of the literature search process used.

**Figure 2 medicina-57-00201-f002:**
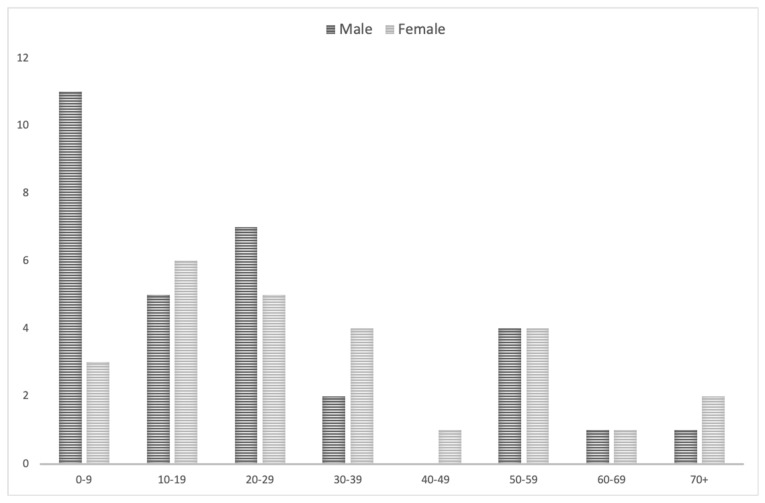
Distribution of patients with de novo arteriovenous malformations according to gender and decade of age at the time of diagnosis. Abscissa: age group (decade of life) for male (dark grey) and female (light grey) patients. Ordinate: number of patients included.

**Figure 3 medicina-57-00201-f003:**
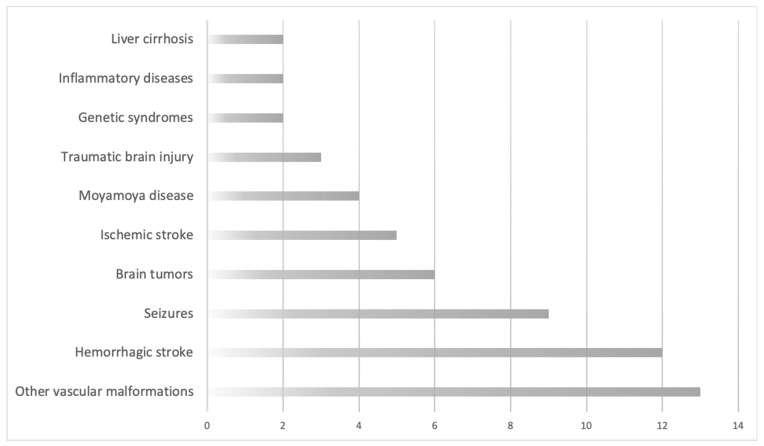
Pathologies allegedly associated with de novo arteriovenous malformations, distributed in ascending order of frequency. Abscissa: total number of patients. Ordinate: associated pathology.

**Figure 4 medicina-57-00201-f004:**
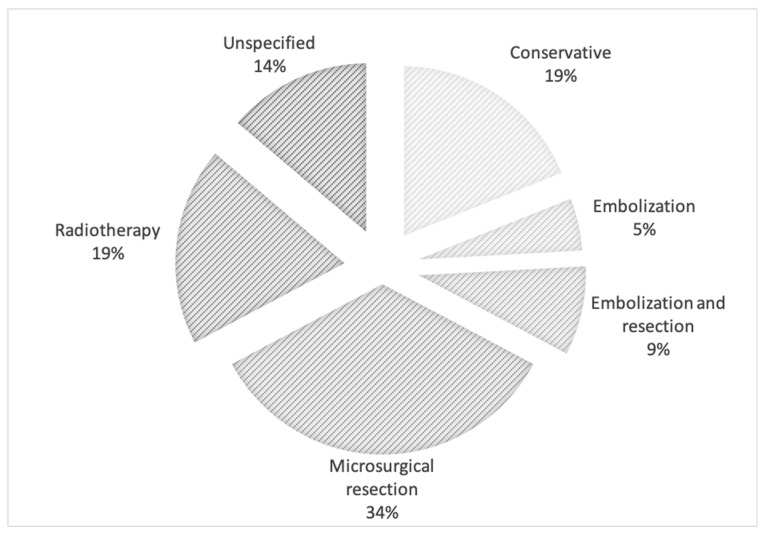
Treatment methods utilized for de novo arteriovenous malformations.

**Figure 5 medicina-57-00201-f005:**
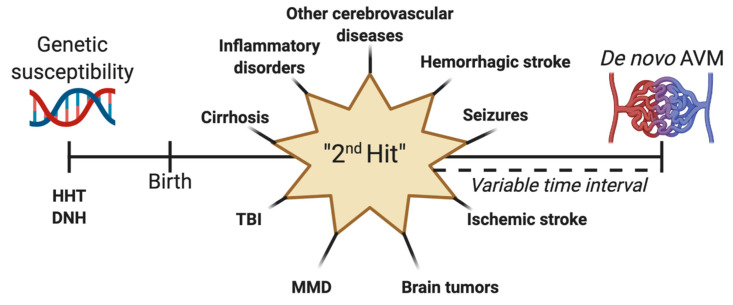
Schematization of the “second hit” theory for the development of de novo arteriovenous malformations (AVM). Genetic susceptibility can be represented by syndromes such as hereditary hemorrhagic telangiectasia (HHT) or diffuse neonatal hemangiomatosis (DNT). After birth, a second may occur, leading to the formation of an acquired AVM after a variable point in time. This second hit can be a preexisting vascular lesion or malformation, hemorrhagic stroke, seizures, ischemic stroke, brain tumors, Moyamoya disease (MMD), traumatic brain injury (TBI), cirrhosis, or inflammatory lesions.

**Table 1 medicina-57-00201-t001:** Illustrates each individual case reported, categorized according to the underlying pathology.

Authors	Sex	Age (yrs.)—De Novo AVM	Initial Symptoms	Imaging (−)	Associated Pathology	Symptoms—De Novo	Imaging (+)	Diagnosis Interval (yrs.)	AVM location	Treatment	Outcome
Akimoto et al., 2003	F	27	Intraventricular hemorrhage (10 yrs. old)	CTA DSA	Splenium AVM & L Occipital AVM − ruptured + resected	Headache, R sided numbness	CTA DSA MRI	17	Cingulate gyrus & corpus callosum	MS	Favorable
Alvarez et al., 2012	M	8	Seizures	CT MRI	Giant R T-O cavernoma	ASX	MRI DSA	2	Pineal	C	Favorable
Bai et al., 2012	M	7	Generalized tonic-clinic seizure	MRI DSA	Dural AVF, embolization; multiple dAVFs simultaneous with AVM	Seizure relapse	MRI DSA	4	L medial Occipital	E	Favorable
de Oliveira et al., 2020	F	16	Headache	MRI DSA	Ruptured R occipital cavernoma, operated	Headache	MRI DSA	10	R Temporal	E+MS	Favorable
Friedman et al., 2000	M	68	Vertigo and vomiting	CT DSA MRI	R tentorial DAVF, treated via SRS and embolization	ASX; 2 years later (age 70)-vertigo and nausea	CT DSA MRI	7	Sup. Vermis	SRS	Favorable
Kawashima et al., 2019	F	29	Cerebellar hemorrhage, ataxia	CT DSA MRI	R cerebellar AVM, ruptured, treated with SRS	headache, vomiting, and worsened ataxia	CT DSA MRI	20	R cerebellar, ectopic	E+MS	Favorable
Lopez-Rivera et al., 2020	M	6	Midbrain hemorrhage	MRI DSA	Midbrain cavernoma, ruptured	Headaches	MRI DSA	2	R Thalamus	C	N/S
Nagm et al., 2015	F	23	Severe headache, seizures	DSA	R frontal AVM, E+MS; ectopic recurrence, reoperated	ASX	MRI DSA	18	R Frontal, ectopic	C	N/S
Nussbaum et al., 1998	M	34	Headache, vertigo, blurred vision	DSA	R cerebellar AVM, C treatment	Recurrence of headache and vertigo	DSA	10	R cerebellar, ectopic	E	Favorable
Rodriguez-Arias, 2000	F	11	Seizures	CT DSA MRI	R parietal AVM, treated via SRS	ASX	MRI DSA	2	R Parietal (medial)	SRS	Favorable
Shi et al., 2018	M	72	Gait instability	CTA DSA	L Frontal AVM (ruptured), resected	L Hemianopsia	CTA DSA	11	R Temporooccipital	MS	Favorable
Shidoh et al., 2016	F	73	ASX	DSA	Unruptured R MCA aneurysm, clipped	ASX	MRI DSA	5	R Parietal (postcentral)	SRS	Favorable
Torres-Quinones et al., 2019	M	80	Vertigo	CT DSA MRI	R superior cerebellar aneurysm, hypertension, vermian AVM treated via radiosurgery	Gait instability	MRI DSA	10	L Thalamus	N/S	N/S
Fuse et al., 2001	M	23	L hemiparesis	CT DSA	R frontal cerebral hemorrhage	ASX	MRI DSA	4	R Frontal	MS	Favorable
Isayama et al., 1991	M	20	Subcortical hemorrhage	CT DSA	L temporal subcortical hemorrhage, evacuated surgically	Second L temporal hemorrhage	DSA	2	L Temporal	MS	N/S
Jeffree et al., 2009	M	18	Headache, facial droop, speech deficit, vomiting, LOC	CT DSA MRI	L Parietal hematoma	Facial droop, aphasia, vomiting, LOC	CT DSA	10	R Parietooccipital	MS	Favorable
Jeffree et al., 2009	M	15	N/S	DSA	R temporo-parietal IPH	ASX	DSA	5	R Temporoparietal	MS	Favorable
Jeffree et al., 2009	M	5	Irritability, poor feeding, vomiting and jaundice	TUS CT MRI	R temporo-parietal IPH at 17 days after birth	Vomiting, episodes of LOC	CT DSA	5	R Sylvian and basal ganglia	SRS	Unfavorable, died 3 months later
Lv et al., 2018	F	22	IPH	DSA	Hemorrhage from N/S cause	Seizures and headaches	MRI DSA	4	R Parietal	N/S	N/S
Mendelow et al., 1987	N/S	N/S	SAH	Angio.	SAH of N/S origin	SAH	Angio.	14	R Occipital	C	N/S
Miyasaka et al., 2003	F	58	Headache, vomiting, L hemiparesis	CT Angio.	R parietal IPH	Severe headache, vomiting	CT DSA MRI	8	R Frontal, R Parietal, L Occipital	SRS	N/S
Morioka et al., 1988	M	23	SAH, LOC	Angio.	SAH of N/S origin	Generalized seizures, postictal R hemiparesis	CT Angio.	7	L Frontal	MS	Favorable
Nakamura et al., 2016	M	53	L frontal IPH	MRI DSA	IPH, evacuated; local implantation of MSCs producing GLP-1	Aphasia, focal seizures, new L frontal cerebral hemorrhage	CTA DSA	3	L Frontal	MS	Favorable
Peeters, 1982	M	26	SAH	Angio.	SAH of N/S origin	Seizures	Angio.	23	R Frontal	N/S	N/S
Porter and Bull, 1969	M	24	SAH	Angio.	SAH of N/S origin	SAH	Angio.	2	R Occipital	N/S	N/S
Dogan et al., 2019	M	25	Seizures, recurrent	MRI	R parietal parafalcine arachnoid cyst, seizures	Recurrent seizures	MRI DSA	14	L Frontal	C	N/S
Kilbourn et al., 2014	M	18	N/S	CT MRI	HCP, absence seizures, autism	Headache, vomiting followed by LOC	CTA DSA MRI	17	Pons	C	N/S
Markham et al., 2015	M	4	Seizures	MRI	Seizures	Severe headache	MRI DSA	3	L Temporal	MS	N/S
Neil et al., 2014	M	24	Seizures	MRI	Epilepsy and head trauma	Seizures	MRI DSA	9	L Parietal	E+MS	Favorable
Ozsarac et al., 2012	M	50	Epilepsy	Angio.	Generalized tonic-clonic seizures	AH (Pink Floyd’s “Another Brick in the Wall”)	CTA	25	L Temporoparietal	C	N/S
Stevens et al., 2009	F	9	Seizures, developmental delay	MRI	SBH (L temporo-occipital)	Seizures, behav. change, ataxia, and aphasia	CT DSA MRI	3	L Parietooccipital	SRS	N/S
Wu et al., 2014	M	7	Fever, convulsions	MRI	Febrile seizure	Seizures, L visual scotoma	MRI DSA	4	R Occipital	MS	Favorable
Yeo et al., 2014	F	16	Single episode of seizures	MRI	Epileptic seizure of N/S cause	Intermittent headaches	MRI	9	L Temporal	MS	N/S
Yeo et al., 2014	M	7	Seizures	MRI	Epilepsy	Recurrent seizures	MRI DSA	6	L cerebellar	SRS	N/S
Bennet et al., 2016	F	45	headache, vertigo, nausea, vomiting	CT MRI	Vermian hemangioblastoma	headaches, dizziness	CTA DSA MRI	1	L cerebellar (posterior)	MS	Favorable
Harris et al., 2000	M	57	headaches, confusion, L sided weakness and numbness	CT carotid Doppler Angio.	R carotid stenosis, malignant R thalamic astrocytoma	Nausea, vomiting, weight loss	MRI Angio.	0,25	R Thalamus	Biopsy, SRS	Unfavorable, died 4 months later
Koch et al., 2016	F	24	developmental regression	MRI	4th ventricle ependymoma, HCP, treatment with surgery + radiotherapy	Partial complex seizure	MRI DSA	23	L Choroidal	SRS	N/S
Lo Presti et al., 2018	F	67	Headaches, L frontal lump	CT	Recurrent meningioma, operated, acrylic cranioplasty	L pulsatile tinnitus	MRI DSA	7	L Frontal	MS	Favorable
Mathon et al., 2013	M	9	N/S	MRI	4th ventricle MB, operated, chemo+radio	ASX	MRI DSA	4	R Sylvian	E+MS	N/S
McKinney et al., 2008	F	58	Transient L-sided weakness	MRI	L thalamic hemorrhage; anaplastic ODG in the same region	L sided weakness, dysarthria, L visual field defect	MRI DSA	3	R Frontoparietal	MS	Unfavorable, died on postop day 6
Morales-Valero et al., 2014	M	56	TIA	DSA	TIA, cause N/S	Transient neurological event	DSA	14	L Frontal	N/S	N/S
Ozawa et al., 1998	M	39	Headache, vomiting, weakness in L arm	CT DSA MRI	Thrombosis of SSS and R TS	ASX	MRI DSA	2	R Parietal	E+MS	Favorable
Pabaney et al., 2016	F	52	R hemiplegia and severe dysarthria	CTA MRI	Acute L frontal ischemic stroke	Generalized tonic-clonic seizure, L frontal IPH	CTA DSA	8	L Frontal	MS	N/S
Santos et al., 2018	M	7	Mild L hemiparesis	MRI DSA	R PComA aneurysm, R hemispheric stroke;	ASX	DSA	6	R Thalamus	C	N/S
Shi et al., 2018	F	33	Headache	MRV	L TS thrombosis	ASX	MRI DSA	2	L Temporooccipital	C	Favorable
Fujimura et al., 2014	F	14	Weakness in upper limbs	MRI	MMD, bilateral revascularization	ASX	MRI DSA	4	R Occipital	C	Favorable
Noh et al., 2014	F	15	TIA on R side	MRI DSA	MMD	ASX	DSA	8	R Frontal interhemispheric	C	Favorable
O’Shaughnessy, 2005	M	6	Infarction, dysphasia, R hemiparesis	MRI	SS-disease, MMD, L frontal cerebral infarction	Mild dysphasia, mild R hemiparesis	MRI DSA	3	R Sylvian	MS	Favorable
Schmidt et al., 1996	M	11	R arm and leg weakness	CT Angio.	MMD, infarction	ASX	MRI DSA SPECT	8	L Parietal	N/S	N/S
Gonzalez et al., 2005	F	7	Head trauma	MRI	Traffic accident, L frontal hemorrhage	Unremitting seizures	MRI DSA	4	R Temporal	SRS	Favorable
Krayenbuhl, 1977	M	12	Headache, vomiting, stiffness of the neck	Angio.	SAH of N/S origin, from head trauma	Headache, LOC, meningeal irritation	Angio.	9	R Frontal	MS	Favorable
Miller et al., 2014	F	12	Head trauma	CT	Severe TBI, L SDH; cranioplasty	Seizures, increased frequency	MRI DSA	11	L Parietal	MS	Favorable
Shimoda et al., 2015	M	5	ASX—screening for HHT	MRI	HHT	Recurrent epistaxis, thunderclap headache	MRI DSA	4	R Parietal	E+MS	Favorable
Song et al., 2007	F	2,5	Congestive heart failure	MRI	Cardiomegaly and CHF; hepatic hemangiomas	Intermittent downward gaze, gait instability, DD	MRI DSA	2,5	L CPA	E	Initially favorable; died at the age of 6 due to cerebral hemorrhage
Bulsara et al., 2002	F	32	Intermittent monocular vision loss, ataxia, diplopia, ptosis, blurred vision, aphasia, gait disturbance	DSA MRI PET	Inflammation/demyelinating lesion in brainstem and diencephalon	Severe headache, nausea, vomiting, ruptured AVM	CTA DSA MRI	6	R Temporal	MS	Favorable
Mahajan et al., 2009	F	30	Facial nerve palsy	MRI	Bell’s palsy	Migraines, R-sided facial weakness, aphasia	CT DSA MRI	14	L Frontoparietal	N/S	N/S
Gondar et al., 2019	F	57	Mild head trauma	MRI	Cirrhosis, R pre-central DVA, TBI after alcohol poisoning	Gait imbalance dysarthria, L hemiparesis	CT DSA MRI	2	R Frontal precentral	SRS	Unfavorable, died 2 weeks later due to PE
Morales-Valero et al., 2014	F	35	Confusion	MRI	Liver cirrhosis, PSE	IPH	MRI DSA	4	L Parietooccipital	N/S	N/S

Reported cases of de novo AVMs in pediatric and adult patients, arranged according to probable correlated pathology. All patients included had an initial negative imaging study for the vascular malformation to have been labeled as ‘de novo.’ Imaging (−) reflects the initial negative study that did not show the presence of the AVM, whereas Imaging (+) signifies the study that revealed the later-diagnosed AVM. An ectopic label for the location of the de novo AVM denotes a lesion in the same general area as a previous lesion, but outside of the boundaries of that lesion. Abbreviations: AH, auditory hallucinations; Angio., angiography; AVM, arteriovenous malformation; ASX, asymptomatic; behav. change, behavioral change; C, conservative; chemo+radio, chemotherapy and radiotherapy; CHF, congestive heart failure; CPA, cerebellopontine angle; CT, computed tomography; CTA, computed tomography angiography; dAVF, dural arteriovenous fistula; DD, developmental delay; DSA, digital subtraction angiography; E, embolization; E+MS, embolization followed by surgical resection; GLP-1, glucagon-like peptide 1; HCP, hydrocephalus; HHT, hereditary hemorrhagic telangiectasia; IPH, intraparenchymal hemorrhage; L, Left; LOC, loss of consciousness; MB, medulloblastoma; MCA, middle cerebral artery; MMD, Moyamoya disease; MRI, magnetic resonance imaging; MRV, magnetic resonance venography; MS, microsurgical resection; MSC, mesenchymal stem cells; N/S, not specified; ODG, oligodendroglioma; PComA, posterior communicant artery; PE, pulmonary embolism; PSE, portosystemic encephalopathy; R, right; SAH, subarachnoid hemorrhage; SBH, subcortical band heterotopia; SDH, subdural hematoma; SRS, stereotactic radiosurgery; SS-disease, sickle-cell disease; SSS, superior sagittal sinus; TBI, traumatic brain injury; TFUS, transfontanellar ultrasonography; TIA, transient ischemic attack; TS, transverse sinus; yrs., years.

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
