# Peer review of "‘De Novo’ Brain AVMs—Hypotheses for Development and a Systematic Review of Reported Cases"

_medicina, 2021, doi:10.3390/medicina57030201_

Round 1

Reviewer 1 Report

This is an interesting paper that established a hypothesis about development of de novo brain AVMs.  Followings are suggestions that may increase the quality of the paper.

Introduction:

  1. For general readers, it would be good to begin with defining the medical term of “AVMs” , followed by clinical features and outcome of brain AVM in the Introduction section.
  2. Also describe about “familial” and “sporadic” brain AVMs. Sporadic form of bAVMs cover about 95% of bAVMs while familial bAVMs are about 5%. Familial bAVMs are mostly associated with HHT. 
  3. “The topography of the arteriovenous lesions in HHT, which favor regions prone to repetitive damage and repair such as the face and lips, suggests that regional angiogenic and inflammatory factors play an important role. “ – this is not a general view in HHT community.
  4. “conditional knock out of either ENG or ALK1 resulted in arteriovenous malformations only when de novo angiogenesis was stimulated by VEGF in mice [30, 79].” – reference 30 and 79 showed “wound” was necessary to induce skin AVMs in adult mice.

VEGF can mimic the wound effect on AVM development – [PMID: 24957885].

  1. It will be worth mentioning the recent papers describing that somatic mutations on KRAS gene have been associated with about 50% of sporadic bAVMs.

Discussion

  1. Line 221 “entrance radiation originating from SRS-treated vascular lesions either facilitated or triggered the development of de novo AVMs [41, 68, 104].” – radiation not only induce inflammation / angiogenesis, but also induce genetic mutations on crucial genes involved in bAVM development such as HHT genes and KRAS.
  2. Line 431 “Two genes that have been linked to the development of HHT are the ENG gene, which is correlated with the more severe type HHT1, and the ACVRL1 or ALK1 gene, which is associated with a more mild phenotype (HHT2) [38, 58].” – HHT1 patients have higher prevalence of developing bAVMs compared to HHT2 patients, but the severity would be comparable.

Author Response

Esteemed editor and reviewers,

We, the authors, would like to express our gratitude for your kind remarks and suggestions for our manuscript. We hope that the changes made are to your expectations. If, however, you would like further changes, we are more than happy to comply. Below, you will find a point-by-point response to each suggestion (in italic), according to the reviewer.

Reviewer 1

This is an interesting paper that established a hypothesis about development of de novo brain AVMs.  Followings are suggestions that may increase the quality of the paper.

Introduction:

  1. For general readers, it would be good to begin with defining the medical term of “AVMs” , followed by clinical features and outcome of brain AVM in the Introduction section.

Response: We are grateful for your remarks and suggestions. We have written a paragraph (lines 88-102) on the general aspects of brain AVMs (definition, clinical features and treatment methods).

  1. Also describe about “familial” and “sporadic” brain AVMs. Sporadic form of bAVMs cover about 95% of bAVMs while familial bAVMs are about 5%. Familial bAVMs are mostly associated with HHT.

Response: We have added this detail in the first phrase of the second paragraph (lines 103-105).

  1. “The topography of the arteriovenous lesions in HHT, which favor regions prone to repetitive damage and repair such as the face and lips, suggests that regional angiogenic and inflammatory factors play an important role. “ – this is not a general view in HHT community.

Response: This phrase was deleted (line 116).

  1. “conditional knock out of either ENG or ALK1 resulted in arteriovenous malformations only when de novo angiogenesis was stimulated by VEGF in mice [30, 79].” – reference 30 and 79 showed “wound” was necessary to induce skin AVMs in adult mice. VEGF can mimic the wound effect on AVM development – [PMID: 24957885].

Response: We have rephrased this sentence, discussing how wounding was necessary for skin AVM development (line 118). We have also added the suggested reference and explained that VEGF can mimic the wounding effect (lines 119-120). 

  1. It will be worth mentioning the recent papers describing that somatic mutations on KRAS gene have been associated with about 50% of sporadic bAVMs.

Response: A brief phrase on the frequency and effects of the KRAS mutation in AVMs was added (line 121-123).

Discussion

  1. Line 221 “entrance radiation originating from SRS-treated vascular lesions either facilitated or triggered the development of de novo AVMs [41, 68, 104].” – radiation not only induce inflammation / angiogenesis, but also induce genetic mutations on crucial genes involved in bAVM development such as HHT genes and KRAS.

Response: We have added a sentence on the possibility that radiation can trigger mutations in AVM-related genes (lines 499-500)

  1. Line 431 “Two genes that have been linked to the development of HHT are the ENG gene, which is correlated with the more severe type HHT1, and the ACVRL1 or ALK1 gene, which is associated with a more mild phenotype (HHT2) [38, 58].” – HHT1 patients have higher prevalence of developing bAVMs compared to HHT2 patients, but the severity would be comparable.

Response: We have rephrased the abovementioned sentence, noting the higher prevalence of brain AVMs in HHT1 (lines 1167-1169).

Again, we would like to thank you for your patience and valuable suggestions..

Reviewer 2 Report

THe authors present a very comprehensive review of de novo AVMs with insightful discussion on pathophysiology.  The paper is well referenced and the discussion is helpful to clinicians and those researching pathobiology of AVM.

Author Response

Esteemed editor and reviewers,

We, the authors, would like to express our gratitude for your kind remarks and suggestions for our manuscript. We hope that the changes made are to your expectations. If, however, you would like further changes, we are more than happy to comply. Below, you will find a point-by-point response to each suggestion (in italic), according to the reviewer.

Reviewer 2:

The authors present a very comprehensive review of de novo AVMs with insightful discussion on pathophysiology.  The paper is well referenced and the discussion is helpful to clinicians and those researching pathobiology of AVM.

Response: We sincerely thank you for your kind remarks. We hope that you will agree with the modifications made for the revision of our manuscript.

Reviewer 3 Report

The authors described "De Novo" brain AVMs through a systematic literature review. Overall, it has been very well written and extensively analyzed. I recommend be accepted for a publication after minor revisions.

Two minor issues:

(1) it seems unknown for the genetic predisposition in many of those reported cases; therefore, the "true" "De Novo" probably is a result from various types of "2nd hit". I would suggest authors incorporate this possibility in the abstract.

(2) For many of known genetic predispositions, there are somatic sporadic mutations with very low frequencies in lesions, which really raise a concern of their actual functionalities in AVMs development, at least for me.  "2nd hit" is a currently reasonable explanation to accommodate this low frequency mutation dilemma in patients.  These limitations shall be briefly discussed. 

Author Response

Esteemed editor and reviewers,

We, the authors, would like to express our gratitude for your kind remarks and suggestions for our manuscript. We hope that the changes made are to your expectations. If, however, you would like further changes, we are more than happy to comply. Below, you will find a point-by-point response to each suggestion (in italic), according to the reviewer.

Reviewer 3:

The authors described "De Novo" brain AVMs through a systematic literature review. Overall, it has been very well written and extensively analyzed. I recommend be accepted for a publication after minor revisions.

Two minor issues:

(1) it seems unknown for the genetic predisposition in many of those reported cases; therefore, the "true" "De Novo" probably is a result from various types of "2nd hit". I would suggest authors incorporate this possibility in the abstract.

Response: We thank you kindly for your remarks. As per your suggestion, we have included the ‘second hit’ hypothesis in the abstract (lines 81-82). However, this has raised the abstract word count to 275. Should a shortening of the abstract be required to below 250 words, we will comply.

(2) For many of known genetic predispositions, there are somatic sporadic mutations with very low frequencies in lesions, which really raise a concern of their actual functionalities in AVMs development, at least for me.  "2nd hit" is a currently reasonable explanation to accommodate this low frequency mutation dilemma in patients.  These limitations shall be briefly discussed.

Response: This is a very valuable point that we have included in the paragraph discussing the study limitations (lines 1379-1382). Should you consider this addition too brief, we will be more than willing to expand it.

Again, we would like to thank you for your patience and valuable suggestions.